# The Transformative Role of Peer Learning Projects in 21st Century Schools—Achievements from Five Portuguese Educational Institutions

Ana Raquel Carvalho * and Carlos Santos

Digimedia, Department of Communication and Art, University of Aveiro, 3810-193 Aveiro, Portugal;
carlossantos@ua.pt
* Correspondence: raquelcabralc@ua.pt

**Abstract:** Rethinking the role of education in the 21st century implies acknowledging the power of learning and the urgency of making learning provision more meaningful, inclusive, and student-centred, which assumes particular importance when learner disengagement is still a global issue in elementary and secondary education. Rooted in social constructivism principles, peer learning is a learner-centred approach that facilitates the development of soft and technical skills, with evidence-based contributions to learners' academic performance under the cognitive, affective, and social dimensions. This study aims to find evidence of the transformative role of peer learning projects in four Portuguese secondary schools and a higher education institution through teachers and peer teacher students' (PTS) perceptions of these projects' purpose, implementation, and impact on the educational community, particularly on PTS. Data were collected by means of a semi-structured in-depth interview and a survey by questionnaire, and content analysis and descriptive statistics were the techniques used. Results show cooperation and interpersonal skills' improvement as major strengths of these projects, whereas the challenges are mostly organisational, e.g., reduced teacher service time and coordination of learners' schedules. Conclusions highlight the potential of peer learning projects to promote pedagogical transformation and innovation in 21st century schools.

**Keywords:** peer learning; innovative practices; pedagogical transformation; 21st century skills; mixed methods

## 1. Introduction

Dealing with 21st century challenges from an educational viewpoint implies considering the power of learning and the key role of learning opportunities in individuals' life journeys [1]. As mentioned by Fullan, Quinn, and McEachen [2] (p. 5): "The new set of crises is forcing humankind to reconsider its relationship to each other, and to the planet and universe: it is essential that we proactively change the world through learning". According to the authors, in order to become deep, learning should be meaningful, happening in relation with others and giving learners the chance to find their purpose, develop abilities, and transform their own realities.

Student dissatisfaction toward school is a global issue yet to be solved [3–8], which might be a symptom of the still existing gap between current educational practice and the needs of 21st century learners and, as stated by González-Rodríguez, Vieira, and Vidal [4] (p. 214), when talking about early school leaving, may be "perceived to be an inefficiency of educational systems". According to Eurostat data on "early school leavers from education and training" referring to 2019 [9], the early school leaving average in the European Union (EU) is 10.2%, close to the goal of 10% or below, settled for EU countries by 2020 [10]. Based on the same data, Portugal is one of the member states registering "the largest reductions ( . . . ) between 2014 and 2019 in the proportion of early leavers" [9] (p. 2), having moved from 17.4% in 2014 to 10.6% in 2019, despite still being slightly above the EU target of 10%.

According to an OECD report on PISA results for Portugal [11] (p. 5), "many students, especially disadvantaged students, hold lower ambitions than would be expected given their academic achievement", which, in the case of Portugal, is still evident, with "one in four high-achieving disadvantaged students", as opposed to "one in thirty high-achieving advantaged students", not expecting to finish tertiary education [11] (p. 5). "Higher risks of social exclusion and lower civic engagement" [12] (p. 26) and "considerable difficulties in the labour market" [9] (p. 1) are reported to be linked to early leaving from education and training [4,12,13], and according to the European Commission [10] (p. 11), there is a correlation between "better educational achievement" and "more active civic participation", which, in the case of countries like Portugal, is reported to be "even more pronounced".

In a world where "scoring high on foundational subjects will not be enough to be competitive" [14] (p. 407), with "entrepreneurial, social and civic competences" [15] (p. 1) being as important as technical skills [16], and with the introduction of collaborative problem solving and learning with digital tools as basic competences for the demands of the upcoming decades [17], transforming education requires deep reflection, flexibility, and improvement of the mechanisms used to identify and assess new ways of learning [18] (p. 146). Voogt, Erstad, Dede, and Mishra [14] (p. 403) referred to the "shortage of creative and innovative workers", and the Council of the European Union [15] (p. 2) reports "a constant high share of teenagers and adults with insufficient basic skills", which, together with the acknowledgement of the growing but not always appropriate integration of digital technologies in contemporary education and learners' lives [7,18,19], might substantiate Erstad's position [3] (p. 76) that "new models of learning and knowledge creation are needed to prepare young people for their future work and citizenship".

The standardisation of learning derived from mass education principles still present in formal education settings, based on which academic achievement strongly relies on testing and reinforces competition [3,8,20]; the prevalence of a "top-down" organisation of "contemporary Western standards" of learning [21] (p. 32); as well as the "pedagogicisation of young people's everyday life" [3] (p. 76), based on excessive focus on academic achievement rather than on learners' interests, backgrounds, and identities, all substantiate Erstad's position [3] (p. 61) that there has been "a lack of understanding about the dynamic processes of learning as part of people's lives". According to the same author, it is essential to "look beyond school" [3] (p. 65) and find in engaging and motivating examples of informal learning environments, such as in community-based initiatives or, as noted by Pereira, Fillol, and Moura [7] (p. 47), simply through young people's informal ways of accessing knowledge "in their leisure time, in digital platforms, in peer communication", the strategies that might contribute to making learning meaningful. Although school is not the only place where learners actually learn, Miño-Puigcercós [6] stresses the central role that school has in learners' lives, which should validate the promotion of alternative ways of engaging learners and favour the strengthening of the bonds between their knowledge, their interests, and the new experiences that school should provide for them.

Rooted in social constructivism principles [22,23], peer learning is a student-centred approach that gives teachers and learners the chance to experience new roles, privileging the human essence of education based on personal interactions [18,24–27]. With evidence-based benefits under the cognitive, but especially affective and social, dimensions [23,28–31], peer learning is in line with Dewey's [32] (p. 46) vision of learning "as an active and constructive process", and Vygotsky's principle of autonomous but scaffolded access to knowledge through the "zone of proximal development" [33] (p. 1), based on which both peer teacher students (PTS) and peer learners (PL) are given the chance to be co-constructors of their learning process [3,25,28,34]. Over the past few decades, improvements in peer learning delivery [30] have contributed to its popularity, particularly in cross-level programs in higher education [22,23,28,29], and justify the presence/emergence of different peer learning varieties, such as peer tutoring, and cooperative learning [24], "the longest established and most intensively researched" [30] (p. 632)—peer teaching [23], peer mentoring [30,35], peer-assisted learning [36], or peer instruction [37]. As stated by

Topping [30] (p. 633), at least 13 organisational dimensions can influence peer learning delivery (e.g., the context; participants' features, such as age, year of study, ability, and role; curriculum content; objectives; assessment of students; and characteristics of the program, such as being voluntary or compulsory). According to a meta-analysis by Balta, Michninov, Balyimez, and Ayaz [28] (p. 67), "learning with and from peers is not always effective", and in order to be successful, peer learning requires planning and structure. Topping [30] (pp. 631–632) distinguishes between the most "archaic perceptions of peer learning", where only the best students were given the chance to transmit their classmates the knowledge instructed by teachers, and more recent trends, according to which "peer helping interaction is qualitatively different" [30] (p. 632), with PTS being cognitively closer to PL and where both "feel equally valuable and worthwhile", being "active participants in the learning process" [30] (p. 643). In order to become deep, learning occurring within the scope of peer helping interaction should result from "reflexive knowledge building" situations, promoted by peer discussion, reflection, and PLs' scaffolded access to knowledge, and not from "knowledge telling" scenarios [22] (p. 3). Simultaneously, by revising information, monitoring, and correcting, PTSs are given the chance to practice cognitive tasks such as summarising, questioning, classifying, and inferencing, considered vital in the development of PTSs' critical thinking and autonomy [29,30]. According to Topping [30] (p. 638), "the greater the differential in ability" between PTS and PL, "the less cognitive conflict and the more scaffolding might be expected".

Within the context of elementary and secondary education, despite the lower number of existing studies, peer learning is considered for its potential to combat early school leaving rates, promoting low-risk environments, and helping learners with behavioural problems to change their attitudes towards school through assuming new roles and restoring their sense of belonging to their educational community [38–40]. However, based on a literature review, more evidence is requested on the effectiveness of peer learning programs [40] as well as on their impact on learners' academic performance [29,35,41], particularly in the case of PTS, since most studies focus on the contributions of peer learning delivery to PL [35,42]. The complexity of isolating variables in the teaching and learning context and identifying cause–effect relationships, as well as the limited design of most studies, very frequently associated with quasi-experimental methods, are pointed out as possible reasons for this [29,41]. Besides this, Marshall, Dobbs-Oates, Kunberger and Greene [35] also report the need for more studies involving peer mentoring programs of different institutions and disciplines, at least in higher education, as well as how important it is for educational institutions to understand the challenges and benefits experienced by PTS when the goal is to run effective peer mentoring programs. Grounded in the circumstances imposed in school year 2019/2020 due to the COVID-19 pandemic, the Portuguese Ministry of Education established a set of guidelines for elementary and secondary schools for school year 2020/2021 [43], with the purpose of supporting learners to restore their learning processes and promote pedagogical innovation. Among the strategies determined in the guidelines is the implementation of cross- and/or same-level peer learning programs by each school, according to which volunteering students become PTSs of their schoolmates, "helping them develop skills, clear doubts, integrate at school, and prepare for assessment tasks" [43] (p. 46). According to the same document [36], the personal, interpersonal, and academic skills development promoted by peer learning, as well as its principles, are in line with what is expected from school and also mirrored in the document released by the Portuguese Ministry of Education with the skills expected from students when they finish compulsory education [44].

Based on learner engagement issues and inequalities, especially in elementary and secondary education in Portugal, on the need for more meaningful and innovative pedagogical approaches as key elements to promote learner engagement, on the potential of peer learning to promote learners' ability to be co-creators of their learning process, but also on the need for more studies focusing on peer learning contributions to PTSs' academic performance as well as involving programs from different institutions and disciplines,

this study aims to find evidence of the strengths and challenges of peer learning delivery in four Portuguese basic and secondary schools and a higher education institution and, through the conclusions drawn, assess its transformative role in 21st century schools. Eight teachers in charge of five peer learning projects and 63 peer teacher students (PTSs) from those projects were surveyed on motivational aspects, human and organisational variables, and participation impact. The sample included teachers and PTSs due to their leading and active roles in the projects. Through a mixed-methods approach, qualitative and quantitative data, collected between December 2018 and January 2019 with the support of a semi-structured in-depth interview and a survey by questionnaire, were analysed by means of content analysis and descriptive statistics. The ultimate goal of the study is to shed light on the transformative role of peer learning programs in schools as well as encourage and support the implementation of similar bottom-up initiatives, improving teachers' experience at school and learners' engagement, inclusiveness, and empowerment.

## 2. Materials and Methods

### 2.1. Purpose of the Study

This multiple case study integrates one of the stages of a research project designed within the scope of educational design research, with the purpose of finding evidence of the strengths and challenges of peer learning delivery in Portuguese schools and, based on the conclusions drawn, informing the development of a prototype of a peer learning program, expected to be an innovative solution to support teaching and learning practice in elementary and secondary education.

Within the scope of this paper, analysis of peer learning delivery in the case of the five projects is expected to answer the following research questions: What was the purpose of the projects?; How were they implemented?; What were the strengths and challenges of project implementation, according to the perceptions of the teachers and peer teacher students who took part in the inquiry?

### 2.2. Sample

Purposeful sampling was the technique used to select the five peer learning projects that constitute this study, based on the following criteria: (i) current/recent implementation in Portuguese educational institutions; (ii) inclusion of cross-level peer learning programs; (iii) voluntary participation of PTS; (iv) evidence of the project outcomes for the learners involved and the educational community. Three of the projects were selected based on news articles reporting their positive impact on the schools in which they were held, one was previously known by the researchers, and another was chosen for convenience due to being implemented by one of the researchers in the corresponding institution. Four of the projects were held in basic and secondary schools and one in a higher education institution. The five institutions were located in four Portuguese districts, namely Vila Nova de Gaia (1 project), Aveiro (2 projects), Santarém (1 project), and Leiria (1 project).

A total of eight teachers and 63 PTS were surveyed, as listed in Table 1. Peer learners (PLs) were not included as part of the study sample, based on the fact that they were not involved in project organisation. As for the teachers, seven worked in basic and secondary schools, among which two assumed non-teaching roles (i.e., a librarian teacher and an educational psychologist), and one lectured in a higher education institution. The commonality between the eight teachers was essentially the fact that all were in charge of the peer learning projects in the corresponding educational institutions and, based on their experience, would be expected to provide valuable insights into the human, pedagogical, and organisational matters of the projects. Regarding PTS, 68.2% were attending upper-secondary education when they participated in the projects: in projects D and E, both scientific–humanistic and scientific–technological courses (42.9%); in project A, only scientific–humanistic courses (9.5%); and in project B, only scientific–technological courses (15.9%). Project C involved former students, in the case of PTS (31.7%), and PLs attending the curricular unit of "Multimedia Laboratory 4" (LabMM4), which integrates the curricu-

lum of a bachelor's degree course on new communication technologies at a Portuguese higher education institution. PLs were attending elementary and/or lower secondary education in four of the projects: elementary education (projects B and E); and lower secondary education (projects A and E). The age of 68.2% of the PTSs ranged from fourteen to eighteen years old (projects A, B, D, and E), and of 31.7% from nineteen to twenty-six years old (project C). The sample included thirty-six female (57.1%) and twenty-seven male (42.9%) PTSs. Among PTSs, the major differences stand between those belonging to 4 of the projects (A, B, D, and E), attending basic and secondary education, and those assuming their role within the context of higher education (project C). This fact is also echoed in the two different age groups represented in the sample, according to which 68.2% of the PTSs are teenagers, as opposed to 31.7% who are young adults. Based on a preliminary data analysis, no significant discrepancy was found between the perceptions of older and younger PTSs regarding similar matters, which substantiates the fact that the impact of PTSs' age group differences was not further assessed within the scope of the study. Teachers and PTSs signed a declaration of informed consent, which, in the case of students under eighteen years old, was done by their legal representatives, to enforce the applicable law regarding participants' confidentiality and anonymity.

**Table 1.** Sample identification.

| Peer Learning Project | Number of Teachers | Number of Peer Teacher Students | Educational Institutions Involved |
|---|---|---|---|
| A | $n = 2$ [T1 and T2] | $n = 6$ | Basic and secondary school |
| B | $n = 2$ [T3 and T4] | $n = 10$ | Basic and secondary school |
| C | $n = 1$ [T5] | $n = 20$ | Higher education institution |
| D | $n = 1$ [T6] | $n = 13$ | Basic and secondary school |
| E | $n = 2$ [T7 and T8] | $n = 14$ | Basic and secondary school |

Own source.

### 2.3. Data Collection Tools and Data Analysis

A semi-structured in-depth interview, to be applied to the teachers, and a survey by questionnaire, to be answered by PTSs, were created and validated for the purpose of this study. Data collection took place from December 2018 to January 2019.

The interview was segmented into three parts, namely the purpose of the projects, human and organisational variables, and results, and was intended to promote teachers' reflection on project implementation aspects as well as on the strengths and challenges resulting from it. Teachers were interviewed individually in the schools in which the projects were developed. The content was recorded, transcribed, and treated under content analysis with the support of qualitative data analysis software WebQDA.

The survey by questionnaire was created on Google Forms and completed by PTSs online. It was segmented into three parts, namely socio-demographic data, initial motivation, and participation in the project, including fifteen closed-ended questions and eight open-ended questions. The survey was intended to identify the profile of the PTSs involved, their motivations to voluntarily integrate the projects, and finally their perceptions of participation in the whole experience. Content from the open-ended questions was analysed under content analysis, also with the support of software WebQDA, and quantitative data were analysed with the support of SPSS by means of descriptive statistics.

Within the scope of this study, teachers' views covered a wider number of variants, whilst PTSs' perspectives were highlighted regarding their reflections about participation in the projects. Although predominantly qualitative, the mixed-methods approach of the study was intended to facilitate triangulation of teachers and PTSs' perceptions of similar variables and, as mentioned by Fraenkel, Wallen, and Hyun [45], regarding the advantages of qualitative and quantitative data combination, contribute to the validity of the study.

Table 2 provides an overview of the interview content in focus, based on the established dimensions, categories, and subcategories resulting from content analysis.

**Table 2.** Overview of interview corpus selected categories and subcategories of analysis.

| Dimension | Categories | Subcategories |
|---|---|---|
| Purpose of the project | Context | Purpose of the project; project description |
| Human and organisational variables | Project design | Who designed; when; target audience |
| | Participants | Teachers; PTS; peer learners |
| | Institutional support | School board; teaching staff |
| | Peer teacher students (PTSs) | More significant challenges faced |
| | Peer learning sessions | Location; schedule; work dynamics |
| | Project management | More significant challenges |
| Results | Project implementation | Strengths; things to be improved; recommendations |

Own source.

Table 3 lists selected open-ended questions from the survey by questionnaire to be paired with categories "peer teacher students" and "project implementation" of the interview, and also establishes a parallel with a closed-ended question of the same survey, selected to promote data triangulation.

**Table 3.** Open-ended and closed-ended question pairing (survey by questionnaire).

| Selected Open-Ended Questions | Selected Closed-Ended Question |
|---|---|
| Q.1 What were the major challenges you had to deal with all over the project? | Q.4 Assess the impact of your participation in the project within the scope of the items listed below. Choose from "really improved" to "regressed". (Items: motivation for learning; self-confidence; sense of belonging to the group; sense of belonging to school; collaborative skills; communication skills; knowledge mastery; leadership skills) |
| Q.2 What were the main benefits resulting from your participation in the project? | |
| Q.3 What advice would you give prospective peer teacher students? | |

Own source.

## 3. Results

### 3.1. Purpose of the Projects

Most projects (A, D, and E) were created with the purpose of "tutoring disadvantaged students and/or providing study support to learners in need". "Preventing early school leaving" was the second most mentioned purpose (B and E). Interestingly, in the case of project B, "early school leaving" was considered regarding PTS attending scientific–technological courses, corresponding to vocational training study programs, and, in the case of project E, regarding PL, as noted in the following comments:

> "We always tried to motivate learners and decrease early school leaving rates, making them feel valued and belonging . . . something students from Scientific-Technological courses report is that they feel a little lessened compared to students from Scientific-Humanistic courses. So, we tried to make them realise things could be different and, by creating bonds with us, teachers, give them the self-confidence they needed for peer learning sessions"

(T3).

Teacher 7 reported the following:

> "In the beginning, the project was considered for supporting learners at risk of early school leaving; the school Director integrated a group that visited several schools and realised many students did not have any support at home to study or do their homework. And this was something that disturbed us and has increased over time"
>
> (T7).

Teacher 5 mentioned two complementary yet opposite purposes, considering both students with learning issues and more proficient learners, namely "increasing the attractiveness level of the subject" and "challenging learners to learn more":

> "In a certain way, it was the high retention rates of LabMM4, probably the second curricular unit with more students having more difficulties ... lacking foundational knowledge of programming and many of them having failed it more than once before. Something had to be done, but obviously without sacrificing the high standards of the subject ... at the same time, another interesting thing to explore was 'what can challenge proficient learners to learn beyond what is addressed in classes and not leave anyone behind' ... "
>
> (T5).

In the case of project B, "accepting an external proposal" was another of the purposes mentioned. In this case, it was a regional Centre of Competence, based on a partnership established with public and private schools, with the intent of subsidising equipment acquisition.

Regarding their nature, four out of the five projects were defined for their humanist and humanitarian nature and two of them also for their motivating and innovative character. Teacher 3 highlighted: "Simply motivating! We were able to create a network within our educational community, involving different stakeholders, and it was inspiring to everyone, thus being so enriching". Teacher 5 associated pedagogical innovation with the new roles assumed by the teachers and PTSs: "The challenge had very much to do with, not losing sight of project design and scientific validation, but giving PTS the chance to take responsibility for as many decisions as possible and implement challenges based on their own ideas".

Overall, the peer learning programs implemented in schools A, B, D, and E were in line with peer tutoring principles, where PTSs helped PLs to restore and improve their skills based on the scaffolded progress provided in all subjects of PLs' curricula, except for project B, according to which the support was centred on a specific subject of PLs' curricula, namely "Robotics". In the case of project C, the peer learning program took place not only within the scope of a specific curricular unit of PLs' syllabus, namely LabMM4, but also included challenge-based and project-based features, according to which each peer learning session (PLS) was implemented based on storytelling and role-playing techniques, and its organisation involved previous collaborative work between PTSs regarding logistical, communication, and pedagogical matters, under the teacher's supervision.

### 3.2. Project Organisation and Implementation

In general, these projects were created based on partnerships, in the case of projects D, B, and E, correspondingly, between the school board, one/two teachers and an existing Nucleus of Citizenship & Personal Development with years of practice and expertise gathered at school or external entities such as a Competence Centre or a Creative Learning Foundation. In the case of project A, the school psychologist, and of project E, the librarian teacher, were, and continued to be, in partnership with other teachers, key elements in creating and implementing the projects. Most projects (B, C, and D) have been implemented since 2016/17 or later and only two (A and E) since 2010/11 or later.

As for institutional support, the encouragement and availability of school boards was unanimously highlighted, and in the case of projects B, C, and D, teachers also underlined

that supporting volunteering initiatives and/or technologically enhanced pedagogical solutions was part of the institutional policy. However, teachers from the five projects reported not having specific service time on their schedules allocated for these peer learning projects, except for two hours of the non-lecturing component (A, D, and E), which was also mentioned as not being sufficient for the management and supervision of the projects. In the case of project E, the librarian teacher transferred six hours allocated for library projects to this peer learning project.

Regarding the way that the remaining teaching staff welcomed the projects, besides curiosity, support, and recognition, there was also some doubt and fear, especially deriving from giving PTSs so much responsibility: "Some teachers immediately supported the idea, but others said I would probably regret giving PTS so much responsibility . . . it was too risky" (T5); "In the beginning there was some fear of whether it was going to work or not, but as soon as results emerged, which was also obvious in learners' attitudes, the community surrendered..." (T7).

As regards recognition of PTSs' volunteering work, projects D and E were the ones where participation in the projects was awarded with a school honor roll diploma at the end of the school year.

Table 4 provides an overall picture of peer learning session (PLS) delivery per project, mostly based on operational variables.

**Table 4.** Peer learning session implementation.

| Variables under Analysis | Indicators | Projects | | | | |
|---|---|---|---|---|---|---|
| | | A | B | C | D | E |
| Location | Specific location at school (e.g., the library, a sports pavilion) | | X | | | X |
| | No specific location at school | X | | X | | |
| | In other schools (elementary schools) | | | | X | |
| Frequency | A weekly session | X | | | X | |
| | Up to 2 weekly sessions | | | | | X |
| | 3/4 sessions per year | | X | X | | |
| Work dynamics | 1 Peer teacher student (PTS)—1 Peer learner (PL) | X | | | | |
| | 1 PTS—2/3 PLs | | X | | X | X |
| | 2/3 PTSs—some PLs | | | X | | |
| Teacher supervision during peer learning sessions (PLS) | On-site (although non-participatory) | | | X | | X |
| | External | X | X | | X | |
| More significant challenges PTSs faced according to teachers' perceptions | Task management in PLS | | X | | | X |
| | Communication issues | | X | | | |
| | Availability and consistency over time | | | X | X | |
| | Impact on PLs' attitudes | X | | | | X |
| | Personal insecurities | | | X | X | X |
| | Logistical issues | | X | | | |

Own source.

Regarding location, the option of having no fixed room for PLS was ultimately connected to preserving the relaxing atmosphere for the sessions or to the sense of openness and proximity between the institution and the students involved: " ... we intend to promote a relaxing atmosphere, where learners feel at ease, so they might search for familiar locations: the library or the recreation room, where they also find desks and in quieter moments can talk ... "

[T1].

As for frequency and duration, most PLS happened on a weekly basis, in PTSs' free time, usually in the afternoon, and lasted between 45 and 90 min. PLS of project C could take up to 2 h 30 min, depending on group performance, and happened at night. In terms of work dynamics, most projects (B, D, and E) relied on small group interactions, with one PTS for two or three PLs.

Regarding teacher supervision, in PLS, teachers were usually not present, and those on-site took a non-participatory though supportive role. According to the teachers, the most significant challenges that PTSs had to deal with were personal insecurities, impact on PLs' attitudes, and availability over time.

*3.3. Outcomes of Project Implementation*

3.3.1. Strengths and Challenges

Based on teachers' and PTSs' answers to the open-ended questions on the strengths and challenges deriving from participation in and implementation of the projects, Figure 1 provides an overview of the variables that more teachers and PTS referred to regarding the projects' strengths and the major challenges dealt with.

| | Teachers' perceptions | Peer teacher students' (PTS) perceptions |
|---|---|---|
| **Strengths** | • Development of interpersonal skills; <br> • Learners' engagement and satisfaction; <br> • PTSs' new role at school; <br> • Simplified communication; <br> • Collaboration and open sharing of information; <br> • Learners' integral assessment; <br> • Decrease in prejudice levels; <br> • Project internal and external recognition; <br> • Impact on institutional assessment. | • Improvement of teaching and learning skills; <br> • Personal and academic development; <br> • Enrolment in humanitarian causes; <br> • Interpersonal relationships. |
| **Major challenges** | • Combining PTS and peer learners' (PL) schedules; <br> • Reduced service time allocated for project management and supervision; <br> • Managing teacher interference in guiding PTS. | • Communicating effectively; <br> • Opting for the best pedagogical strategies to clarify PLs' doubts; <br> • Dealing with PLs' attitudes; <br> • Having positive impact on PLs' attitudes. |

**Figure 1.** Teachers and peer teacher students' perceptions of project implementation outcomes (own source).

The following extracts from teachers' comments exemplify some of the indicators included in Figure 1:

"When they work and ask each other for help, when they assume others know it better, they are collaborating with each other."

[T7]

"The whole experience shows that also PTS who did not score so high in tests, here have the opportunity to shine, and they do because they are available, and because of their attitude and commitment. All these components are assessed, so they see their academic results improve."

[T1]

"One of the PTS once told me, almost disappointed, he was going to work with a Romani PL . . . after some time, the same PTS reported the close and affectionate bond both were creating . . . I know he will never look at the Romani community the same way . . . "

[T6]

In the case of PTSs' comments, the following extracts stood out:

"It was so enriching . . . helping others and finding new ways of explaining things helped me strengthen my own weaknesses."

[PTS, project C]

"The PL I worked with improved his academic results, started to have study habits and it was good to me as it has helped me improve my self-esteem."

[PTS, project A]

"I got a clearer perception of what teamwork involves."

[PTS, project B]

"I had the chance to create bonds with children and I loved it. I became more organised and it was really nice to feel that an action can change someone's life, including mine."

[PTS, project D]

### 3.3.2. Participation Impact on Peer Teacher Students' Academic Performance

As regards PTSs' perceptions of participation impact on their academic performance, data collected from the selected closed-ended question: Q.4, "Assess the impact of your participation in the project within the scope of the items listed below. Choose from 'really improved' to 'regressed'—focused on variables "motivation for learning"; "self-confidence"; "sense of belonging to the group"; "sense of belonging to school"; "collaborative skills"; "communication skills"; "knowledge mastery"; and "leadership skills".

Results related to the eight variables assessed are shown in Figure 2. According to this figure, "collaborative skills" is the most impacted variable, with the highest score under both options "really improved" (29%) and "improved" (59%). "Sense of belonging to the group" (24%), "to school" (19%), and "motivation for learning" (19%) were the second and third variables most marked as having "really improved". Furthermore, PTSs also considered that their "self-esteem" (59%), "communication skills" (56%), and "motivation for learning (54%) were the variables with the highest scores for having "improved". Inversely, variable "leadership skills" was considered by more PTSs (41%) as not having changed, although 48% still marked it as having "improved" and 8% as having "really improved". Overall, the impact of participation in the projects was significantly positive, with all variables being scored higher for having "improved" over time.

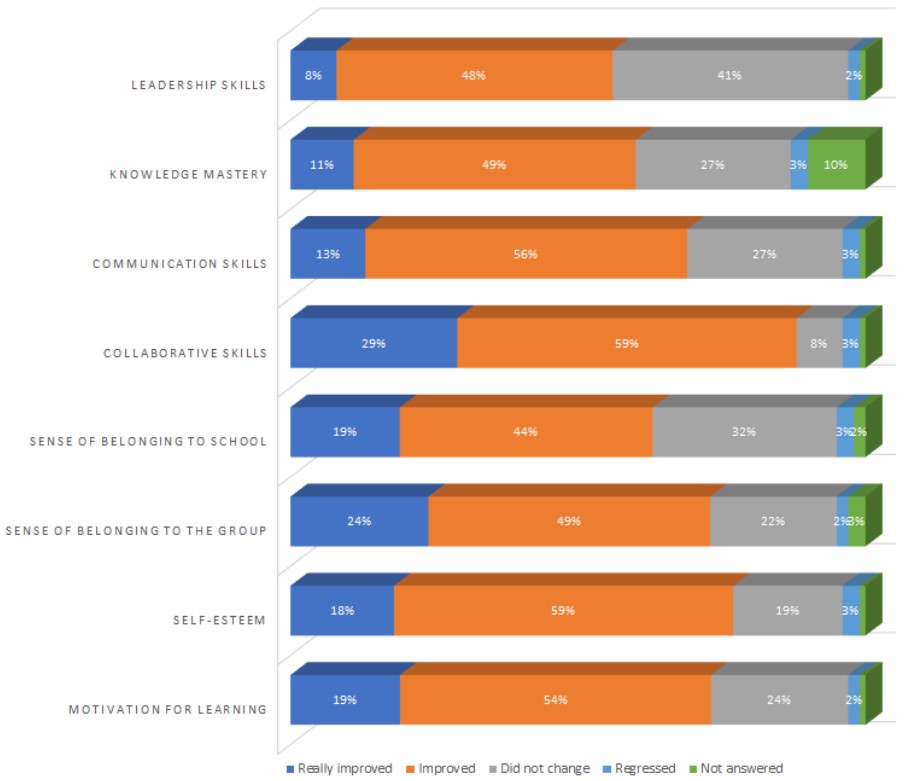

**Figure 2.** Peer teacher students' perceptions of participation impact on their academic performance (own source).

Furthermore, in regard to the above-mentioned data, a binomial test was performed with the support of SPSS based on the following hypothesis:

$$H_0 : p = 50\% \tag{1}$$

$$H_1 : p > 50\% \tag{2}$$

The cut-off point selected was level 2, splitting the sample into two groups:

$$Group\ 1 : \leq\ 2 \tag{3}$$

("did not change"/"regressed")

$$Group\ 2 : \geq\ 2 \tag{4}$$

("really improved"/"improved")

Figure 3 provides the binomial test results and, according to them, only in the case of variables "leadership skills" and "knowledge mastery" were the *p*-value results higher than = 0.05, which means that for the remaining variables, the *p*-value results show statistical significance. Based on this, it can be inferred that there is statistical evidence that, according to PTSs' perceptions, participation in the projects had a significant impact on their academic performance, since all variables assessed, except "leadership skills" and "knowledge mastery", were rated by more than half of the PTSs as having "really improved" or "improved" over time.

### 3.3.3. Recommendations to Prospective Participants of Peer Learning Projects

Finally, recommendations highlighted by the teachers as to requisites for peer learning project implementation focused on the professional skills that teachers require to manage and supervise peer learning projects, namely "commitment, availability, and tolerance";

human and interpersonal skills like "working in multidisciplinary teams", "creating networks", "giving PTS as much responsibility as possible"; and also organisational variables like "planning short PLS", "promoting interdisciplinarity", "efficiently managing learners' schedules"; and "having institutional support".

| Variables assessed | | Category | N | Observed Proportion | Test Proportion | Exact Sig (bilateral) |
|---|---|---|---|---|---|---|
| Leadership skills | Group 1 | <= 2 | 28 | 0.44 | 0.50 | 0.450 |
| | Group 2 | > 2 | 35 | 0.56 | | |
| | Total | | 63 | 1.00 | | |
| Knowledge mastery | Group 1 | <= 2 | 25 | 0.40 | 0.50 | 0.130 |
| | Group 2 | > 2 | 38 | 0.60 | | |
| | Total | | 63 | 1.00 | | |
| Communication skills | Group 1 | <= 2 | 20 | 0.32 | 0.50 | 0.005 |
| | Group 2 | > 2 | 43 | 0.68 | | |
| | Total | | 63 | 1.00 | | |
| Collaboration skills | Group 1 | <= 2 | 8 | 0.13 | 0.50 | 0.000 |
| | Group 2 | > 2 | 55 | 0.87 | | |
| | Total | | 63 | 1.00 | | |
| Sense of belonging to school | Group 1 | <= 2 | 23 | 0.37 | 0.50 | 0.043 |
| | Group 2 | > 2 | 40 | 0.63 | | |
| | Total | | 63 | 1.00 | | |
| Sense of belonging to the group | Group 1 | <= 2 | 17 | 0.27 | 0.50 | 0.000 |
| | Group 2 | > 2 | 46 | 0.73 | | |
| | Total | | 63 | 1.00 | | |
| Self-esteem | Group 1 | <= 2 | 15 | 0.24 | 0.50 | 0.000 |
| | Group 2 | > 2 | 48 | 0.76 | | |
| | Total | | 63 | 1.00 | | |
| Motivation for learning | Group 1 | <= 2 | 17 | 0.27 | 0.50 | 0.000 |
| | Group 2 | > 2 | 46 | 0.73 | | |
| | Total | | 63 | 1.00 | | |

**Figure 3.** Binomial test results. (own source).

In the case of recommendations given by PTS to prospective PTS, the majority referred to the importance of personal and behavioural requirements like "being calm and patient", "relaxed", "being committed and available"; as well as humanist and humanitarian aspects like "enjoying helping others" and "focusing on the benefits of participation" to be consistent over time.

## 4. Discussion

Based on the study results, it may be inferred that to be successful, peer learning project-based initiatives depend on integrated coordination between the school board, the teachers involved, and the remaining teaching staff. Other elements of the pedagogical

staff holding a more flexible schedule, such as educational psychologists and/or librarian teachers, are welcomed for the support provided as well as the possible allocation of extra service time for such initiatives, as happened in projects A and E. According to the European Commission's report on "recommended instruction time in compulsory education" [46] (p. 134), "schools may manage up to 25% of the curriculum in a flexible way" and, in this way, develop curricular enrichment activities. As a matter of fact, the major challenges reported by teachers in this study focused on organisational aspects regarding the need for more teacher service time provision for the projects and for better planning to facilitate the management of PTSs' and PLs' schedules. Based on the reported guidelines [46], schools currently have more tools to provide these projects with better conditions, so that peer learning implementation does not solely depend on the volunteering commitment of hardworking teachers.

As stated by some of the teachers, another challenge that they faced was the reluctance of other teachers to believe in learners' capacity to assume a new role that required so much responsibility and autonomy. Bearing in mind the significance of learner-centred approaches in 21st century schools [24,25,27,28,30,31], and the key role of teachers in reducing the gap between learners' needs and current educational practice [3,6,47], it is mandatory that teacher training meets the standards of 21st century educational scenarios [6,7] and supports teachers to (re)signify the meaning of teaching and learning in the academic community [1,18].

According to the study results, most peer learning sessions happened on a regular basis, which, on the one hand, reinforces affective bonds and promotes learners' scaffolded progress, in part also due to the distinguishing role of ICT tools based on their ubiquity, immediacy, and effective promotion of collaboration and communication between PTS and PL [48], but on the other hand requires consistent effort, commitment, and availability over time, which, for some PTSs, may be a challenge, as stated by the teachers. However, based on PTSs' answers, the major challenges reported were related to effectively performing their new role, mostly regarding communication and choice of appropriate pedagogical strategies, and this is in line with the teachers' opinion that dealing with "personal insecurities" was probably PTSs' major challenge. Curiously, by comparing the results, the same PTSs reported "the improvement of teaching and learning skills", when answering the open-ended question on the main benefits of participation, and registered major development of their "collaborative" and "communication skills" (as stated in Figure 2), when answering the selected closed-ended question on participation impact on their academic performance. Only a very small percentage of PTSs reported a regression in most of the skills in focus, having associated PLs' lack of commitment as the main challenge they faced, and the personal development and satisfaction deriving from helping others as the main benefits of the experience. All this may indicate that independently of age group differences, PTSs' perceptions tended to converge and that, despite the demanding tasks of PTSs' new roles, most students realised what effective peer helping interaction requires [29,30] and were able to overcome insecurities, adapt, and assume an active and constructive attitude, as recommended by Binkley, Erstad, Herman, Raizen, Ripley, Miller-Ricci, and Rumbler [17] (p. 32), when authors associate "sophisticated thinking, flexible problem solving, and collaboration and communication skills" as the "new standards for what students should be able to do", "to be successful in work and life" and based on which schools must promote transformation. Simultaneously, and based on Gillie's [24] remarks on how to effectively promote successful cooperative group work, it is essential to provide PTSs with the skills and support that may help them to internalise key elements described by the author, for promoting "successful cooperation" (p. 3), namely ensuring "positive interdependence", based on which learners develop their sense of belonging to the group and influence within the group, "individual accountability" [24] (p. 3), improving PTSs' ability to "actively listen to others, ( . . . ), constructively critiquing the ideas of others, sharing resources, taking turns" (p. 4), effectively promoting "interactions", and finally "group processing" or reflecting on achievements and challenges to be further improved

(p. 4). When researching into the effectiveness of peer learning programs and their layout, it is therefore relevant to assess how PTSs are prepared for the role and how this impacts the way that they perform this role.

Besides this, the reported development of "personal and interpersonal skills", both by teachers and PTSs, as major strengths deriving from participation in the projects, also substantiates the position of authors who state the emotional and social nature of human beings [6,26,49,50] and the critical impact of emotions on cognition [49]. Together with skills like creativity and critical thinking, all these assumptions contribute to a comprehensive perception of what learning means, and, as highlighted by Erstad [3] (pp. 63–64): "By combining 'knowing' and 'becoming' ( . . . ) we open up a more dynamic understanding of learning". Based on this, it may be inferred that, despite including voluntary PTSs who, for that reason, were willing to participate in the projects and were less likely to have engagement issues at school, the five projects gave learners from different course backgrounds the chance to strengthen their affective and social bonds, clearly expressed in PTSs' positive perceptions of participation impact on their sense of belonging to school and to the group, but also on their self-esteem and motivation for learning. These facts show these projects' potential to captivate different learner profiles, including those of students with higher risk of social exclusion and learning engagement issues, giving them the opportunity to feel valuable and to restore their relationship with school [6,38–40], their appreciation for learning, and after all to be used by 21st century schools as a bottom-up pedagogical solution to promote learner engagement and combat early school leaving.

## 5. Conclusions

According to the study results, most of the peer learning projects analysed show similarities regarding their purpose, organisation and implementation features, and impact on the corresponding participants and educational communities, especially when sharing the same educational context, as happened with four of the projects, held in basic/elementary and secondary schools. Coincidently, major differences were reported between these and the project implemented in higher education, specifically regarding the peer learning principles of the program and the role given to PTSs, whose tasks/responsibilities are, within this scope, more student-centred but also require more autonomy and availability over time, which seems to be one of the challenges that PTSs have to cope with in secondary education, especially when such project-based initiatives are extra-curricular. As for the achievements, no significant differences were found between the perceptions of teachers and PTSs' in the five projects, which highlights the unanimity of the benefits reported within this scope.

Based on the study results, its contributions are expected to be of particular interest to researchers, educational leaders, teachers, and the educational community in general. By analysing human, pedagogical, and organisational variables of peer learning projects implemented by different educational institutions, involving learners from distinctive courses and disciplines, the study provides comparative results that are expected not only to consolidate findings but also to add insights into the establishment of guidelines for peer learning delivery [40], particularly in basic and secondary education. Besides this, study results on PTSs' participation impact on their academic performance, based on these students' perceptions of the strengths and challenges experienced and complemented by the teachers' views on the same matters, is expected to expand knowledge of a peer learning area reported in the literature for being "less studied" [35] (p. 1), and contribute to providing clarity on how to effectively organise and implement peer learning programs, bearing in mind not only PLs but also PTSs' gains resulting from it [29,41]. The choice of a mixed-methods approach that combines the richness of qualitative data with the validity provided by quantitative data [45] is also expected to be an alternative to methods reported in the literature for not being the most effective when the purpose is to analyse educational variables and identify cause–effect relationships [29,41]. Contributions of the study are expected to be particularly useful for teachers and educational institutions that might

be interested in using peer learning project-based initiatives as a bottom-up solution to implement more meaningful and innovative pedagogical approaches and promote learner engagement, inclusiveness, and empowerment. It may hence be inferred that, according to the study findings, the peer learning projects in focus have promoted the development of what Voogt, Erstad, Dede, and Mishra [14] (p. 407) call "key areas" of 21st century curriculum, namely "foundational", "meta", and "humanistic knowledge", providing all learners with the opportunity to find their place at school [6], engage "in what matters for them and their communities" [3] (p. 67), and play a transformational role in the schools to which they belonged.

As for the limitations of the study, despite including five educational institutions that, especially in the case of basic and secondary schools, may be considered representative of the corresponding educational context in Portugal, due to the small size of the sample, the findings may not be generalisable. In addition, further investigation would be needed to complement the findings for the evidence of peer learning contributions on learners' academic performance, especially regarding the development of cognitive and metacognitive skills. In terms of methods, including PLs in the study sample and assessing their own perceptions of participation in the projects, particularly regarding the main challenges and benefits resulting from it, would promote triangulation of the data gathered and more robust confirmation of findings. Although it would be impossible to conceive it with all learners included in the sample, using the focus group technique with a reduced but representative number of PTS selected from the five projects would be a valuable way to add in-depth understanding of these learners' perceptions of the matters in focus. Future research on peer learning project-based delivery in similar contexts in other countries, including longitudinal assessment of PTSs' and PLs' satisfaction toward learning, may be a valuable contribution to complementing findings on the challenges and strengths of peer learning delivery and support its widespread use in more educational scenarios. Besides this, based on the pedagogical innovativeness of peer learning, research on the strengths and challenges of peer learning deployment during the COVID-19 pandemic, either as a blended learning or a distance learning solution, would add valuable insights into the perceptions of its effectiveness when being mediated by digital technologies and into its adaptability potential, not only to diverse contexts and audiences but also to new and challenging educational scenarios. Simultaneously, implementing studies involving different educational institutions and learner profiles, based not only on participants' perceptions but also on complementary data (e.g., input from learners' interactions within the scope of the peer learning tasks) would promote the conditions to assess gains more effectively to the learners involved in peer learning programs under the cognitive, affective, and social dimensions.

**Author Contributions:** Conceptualization, A.R.C. and C.S.; methodology, A.R.C. and C.S.; software, A.R.C. and C.S.; validation, C.S. and a group of experts.; formal analysis, A.R.C. and C.S.; investigation, A.R.C. and C.S.; resources, A.R.C. and C.S.; data curation, A.R.C. and C.S.; writing—original draft preparation, A.R.C. and C.S.; writing—review and editing, A.R.C. and C.S.; visualization, A.R.C. and C.S.; supervision, C.S.; project administration, C.S. funding acquisition, A.R.C. All authors have read and agreed to the published version of the manuscript.

**Funding:** This research was funded by The Portuguese National Funding Agency for Science Research and Technology ("Fundação para a Ciência e a Tecnologia – FCT") under Grant reference SFRH/BD/146606/2019. The APC was funded by Digimedia – Digital Media and Interaction Research Centre, University of Aveiro.

**Institutional Review Board Statement:** Ethical review and approval were waived for this study, after assessment of the Ethics Committee of the University of Aveiro, due to the fact that the nature, scope, context, and purpose of the data collected and processed were decided not to be likely to result in a high risk to the rights and freedoms of the persons involved. The study was implemented in compliance with the applicable law regarding data protection as well as participants' confidentiality and anonymity.

**Informed Consent Statement:** Informed consent was obtained from all subjects involved in the study.

**Data Availability Statement:** The authors ensure that the data shared are in accordance with consent provided by participants on the use of confidential data. However, the data supporting reported results cannot be made available at the moment of this article publication, based on data availability restrictions regarding the research project within the scope of which the present study was implemented, imposed until the project has finished. Notwithstanding, access to the data supporting study results is expected to be possible from the first quarter of 2022 onwards.

**Acknowledgments:** The support of the Portuguese National Funding Agency for Science Research and Technology ("Fundação para a Ciência e a Tecnologia – FCT") as well as of Digimedia – Digital Media and Interaction Research Centre is gratefully acknowledged. We would also like to thank the schools that integrated this study for their receptiveness. The presented data were originally collected in Portuguese and this is a free proposed translation by the authors.

**Conflicts of Interest:** The authors declare no conflict of interest. The funders had no role in the design of the study; in the collection, analyses, or interpretation of data; in the writing of the manuscript, or in the decision to publish the results.

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
