# Peer review of "The Transformative Role of Peer Learning Projects in 21st Century Schools—Achievements from Five Portuguese Educational Institutions"

_education, doi:10.3390/educsci11050196_

Round 1

Reviewer 1 Report

This study is interesting in that it took the  well known strategy of peer teaching/learning and investigated a range perspectives across year groups and schools. The mixed method design was an appropriate choice, as the qualitative statements and findings could be backed up by descriptive statistics. The paper is well written with only a few minor English errors. One is the incorrect use of the word 'inquired' in several places. For example, in line 152  it is stated '63 PTS were inquired'. This is incorrect English.  Instead you could say 'were investigated', or 'were surveyed' or 'took part in the inquiry' depending on the intent of the statement.

While the literature review dealt with early school leaving, there was little further discussion of this aspect in relation to the findings. Further, while the impact of Covid 19 on Education was mentioned in the beginning, the discussion did not indicate how peer learning might address that.

Overall though,  the study and the paper  are well presented and worthy of publication.

Author Response

"Please see attachment."

Reviewer 2 Report

Thank you for the opportunity to revise the paper: “The transformative role of peer learning projects in 21st century schools – achievements from five Portuguese educational institutions”

I believe this paper addresses an interesting topic, it is well written, the arguments in general are sound and follow a logic, and the methodology is well implemented. As constructive comments, I would mention the following:

I think the introduction does a good job at presenting the importance of the topic and the goal of the paper. However, it is not very develop how the paper builds on previous literature, both in terms to fill an existing gap and even more importantly to develop the arguments that constitute the core of the theoretical foundation. There is a nice paragraph building on constructivism and linking it to peer learning, but not much more. While this would be normal in a regular introduction in which the section is followed by a Literature Review section, this is not the case in this paper in which there is no literature review section, and therefore, either the authors include such section, or they need to cover it more comprehensively in the introduction.

Could it be possible to provide some descriptive statistics of the subjects in the interviews and survey? And to what extent differences in the subjects could affect the results obtained?

Similarly, the contributions of the paper seems also relatively short, and not well grounded in theory. Similar to my first comment, it would be nice if the authors could link more the paper to an existing scholarly debate and show how their paper expands our knowledge in this domain.

I also think the authors could develop further the section of limitations (for example, to what extent the schools covered are representative of Portugal; which other techniques or data could have allowed a deeper and more robust confirmation of findings), and most importantly, the one about future avenues for further research that arise from this paper, as this would be very helpful to other researchers.

I wish the authors good luck in their project!

Round 2

Reviewer 2 Report

No more comments